# Duration of Immersion and Type of Immersion Solution Distort the Outcome of Push-Out Bond Strength Testing Protocols

**DOI:** 10.3390/ma12182860

**Published:** 2019-09-05

**Authors:** David Donnermeyer, Lena Göbell, Sebastian Bürklein, Till Dammaschke, Edgar Schäfer

**Affiliations:** 1Department of Periodontology and Operative Dentistry, Westphalian Wilhelm-University, 48149 Münster, Germany; 2Central Interdisciplinary Ambulance in the School of Dentistry, Westphalian Wilhelm-University, 48149 Münster, Germany

**Keywords:** endodontic sealer, protocol standardization, push-out bond strength, AH Plus, BioRoot RCS

## Abstract

This study aimed to investigate the influence of immersion duration and the type of immersion solution on the outcome of push-out bond strength (POBS) tests. Root canals of 120 straight single-rooted teeth were instrumented to a diameter of 1.5 mm and irrigated with 5 mL of 3% NaOCl. Four horizontal slices with a thickness of 1 mm were cut, representing the mid-portion of the root. The specimens (n = 480) were irrigated with 17% ethylenediaminetetraacetic acid(EDTA) for 60 seconds, then twice with distilled water (DW) for 30 s each. The canals were filled with either AH Plus (Dentsply Sirona, Konstanz, Germany) or BioRoot RCS (Septodont, St. Maur-des-Fossés, France) (n = 240). Separated into four groups per type of sealer (n = 60), the specimens were incubated at 37 °C covered with gauze moistened in DW or phosphate-buffered saline (PBS) for either one or eight weeks. Dislodgement resistance was measured and POBS was calculated. Statistical analysis was performed using the analysis of variance (ANOVA) test and the Student–Newman–Keuls test (*p* = 0.05). AH Plus showed higher POBS when stored in PBS compared to DW, irrespective of the incubation period (*p* < 0.05). BioRoot RCS displayed higher POBS when stored in DW compared to PBS after eight weeks of incubation (*p* < 0.05). No difference was found after one week of incubation (*p* > 0.05). Irrespective of the sealer or the immersion solution, POBS decreased from one week to eight weeks (*p* < 0.05). Mixed failure modes were found in all groups irrespective of sealer, immersion medium, or immersion period. POBS decreased after a longer incubation time in both immersion solutions. Duration of immersion and the type of immersion solution had a significant impact on the outcome of the POBS testing protocol.

## 1. Introduction

Push-out bond strength (POBS) tests are well established in the laboratory assessment of characteristics of endodontic sealers [1], irrespective of the clinical applicability of the results [2]. Innumerable protocols have been described in past literature, but so far, no protocol standardization has been achieved [2]. Standard protocols, mostly ISO (International Standard Organization) or ANSI/ADA (American National Standards Institute/American Dental Association) specifications, are available for the most common laboratory investigations of sealers, such as flow, setting-time, radiopacity, and several others, but there are none for POBS investigations [2,3]. This creates a difficulty when comparing results of different studies and it also leads to numerous efforts being made to establish new protocols in the hope of creating the optimum accepted model. Collares et al. reported on the influence of different methodological variables on POBS tests [3]. According to their systematic review, POBS values were influenced by country and year of publication, use of human or bovine dentine, tooth portion, type of sealer, core material, obturation technique, and testing-machine velocity, as well as the duration of specimen storage [3]. Accordingly, variables such as the ratio of the plunger diameter and the specimen’s obturation diameter [1,4,5], different canal preparation protocols [2], the use of gutta-percha as a core material versus sealer as the sole obturation material [6], and the use of bovine or human dentine [7] have been reported in the literature as affecting the push-out model. 

Knowledge about the impact of relevant parameters of the experimental setup on the results of push-out bond strength studies is scarce. Up to now, it has been unclear whether different durations of immersion or the use of different immersion solutions exerts any impact on POBS values. Only different periods of specimen incubation at 100% humidity were reported to affect the outcome of POBS tests of calcium silicate-based sealers [8]. Throughout the literature, different storage periods ranging from a few days up to several months have been reported [3]. Furthermore, specimen storage was described as involving incubation at different levels of humidity, at different temperatures, under dry and wet conditions, and in different immersion solutions [3,9]. Moreover, many of these factors were not described precisely enough in the methodological protocols of some studies. Also, no information was found in the literature concerning the type of immersion solution affecting POBS. Due to the different chemical compositions of endodontic sealers (e.g., epoxy resin sealers, zinc oxide eugenol sealers, calcium silicate-based sealers, and many more), it is likely that these sealer types may interact differently with different immersion solutions. For example, in relation to the physical properties of calcium silicate-based sealers, it was reported that these materials were severely affected by the type of immersion solution (distilled water (DW), Hank’s balanced salt solution (HBSS), or Dulbecco’s modified eagle medium (DMEM)) [10]. At the least, calcium silicate-based sealers seemed to be affected by the type of immersion solution in their physical properties such as setting-time and leaching of ions [10]. Furthermore, precipitates of calcium hydroxyapatite were reported after storage in phosphate-buffered saline (PBS), indicating the differences in the interaction of calcium silicate-based materials with contact liquids, as this formation was not reported after storage in DW [11]. 

The abovementioned facts illustrate the complexity of sealer testing protocols, especially of POBS testing protocols, as minor differences in the testing protocol might lead to a distortion of results. Hence, comparing results of different push-out bond strength protocols is hardly possible, and conclusions are limited within a certain study [2]. 

As no sufficient information could be found regarding the effect of type and duration of immersion on the results of POBS protocols, the aim of this study was to evaluate the duration of immersion and the type of immersion solution as possible confounding factors. A calcium silicate- based sealer (BioRoot RCS, Septodont, St. Maur-des-Fossès, France) and an epoxy resin sealer (AH Plus, Dentsply Sirona, Konstanz, Germany) were stored in PBS and DW for one and eight weeks, respectively. The null hypothesis tested was that neither the duration of immersion nor the type of immersion solution would exert an impact on the obtained POBS values. 

## 2. Results

Multifactor ANOVA revealed significant influences for the parameters *sealer* (*p* < 0.05), *duration of immersion* (*p* < 0.05), and *type of immersion* (*p* < 0.05). Also, the interaction between *type of immersion* and *duration of immersion* (*p* < 0.05) and the interaction between *sealer* and *type* of immersion (*p* < 0.05) revealed significant influences on the outcome of POBS values during multifactor ANOVA. The interaction between *duration of immersion* and *sealer* did not reveal significant influences on the outcome of POBS values during multifactor ANOVA (*p* > 0.05). 

All specimens had measurable adhesion to the root dentine and no premature failure occurred. AH Plus showed the highest POBS after one week of immersion in PBS (*p* < 0.05). The POBS of AH Plus significantly decreased after eight weeks of immersion in PBS (*p* < 0.05). POBS of AH Plus stored in DW for one week was significantly lower compared to storage in PBS after one week (*p* < 0.05), and it decreased significantly over the eight week period as well (*p* < 0.05). In general, the POBS of AH Plus was significantly higher than that of BioRoot RCS when comparing the same immersion solutions and immersion periods (*p* < 0.05). The POBS of BioRoot RCS was comparable when stored in DW or PBS for one week (*p* > 0.05). In both immersion solutions, the POBS of BioRoot RCS decreased significantly over the eight week immersion period (*p* < 0.05), while the POBS of BioRoot RCS stored in PBS was significantly lower than when it was stored in DW after eight weeks of immersion (*p* < 0.05), as shown in Table 1.

Irrespective of the sealer used, the immersion medium, and the immersion period, all groups showed mainly mixed failure modes, as shown in Figure 1. Regarding AH Plus, the percentage of adhesive failure modes increased with both immersion media from one week to eight weeks of immersion. For BioRoot RCS, the percentage of adhesive failure modes decreased with both immersion media from one week to eight weeks of immersion. BioRoot RCS specimens immersed in PBS showed a more pronounced increase of cohesive failure modes compared to those immersed in DW.

## 3. Discussion

Push-out bond strength protocols are commonly used to evaluate the adhesion and dislodgement resistance of root canal sealers to the root canal wall and/or a core material [3]. Lately, the interference of immersion solutions on the results of ISO-standardized sealer testing protocols, such as setting time, flow, film thickness, sorption, porosity, and solubility, have been shown for different sealer types [10]. In ISO 6876:2012 [12]—a specification for standard tests on endodontic sealers—DW is proposed for use in, for instance, solubility examination. It has been discussed whether immersion solutions with osmolarities simulating tissue fluids are more appropriate to investigate incorporated materials and how they affect the outcome of sealer testing protocols [10]. The present study was hence designed to figure out the influence of PBS solution on the push-out bond strength of an epoxy resin and a calcium silicate-based sealer compared to DW after different periods of immersion.

AH Plus showed higher POBS than BioRoot RCS after storage in both immersion media and after both immersion times. In both immersion media, POBS decreased for both AH Plus and BioRoot RCS in the one and eight week immersion times. The type of immersion solution exerted different effects on the POBS of AH Plus and BioRoot RCS. Superior POBS of AH Plus compared to BioRoot RCS and other calcium-silicate-based sealers have been reported before [13,14,15]. To the best of our knowledge, no study so far has compared the POBS after different immersion times and storage in different immersion media. AH Plus showed higher POBS when stored in PBS after both immersion times. Over an eight week period, the POBS of AH Plus decreased significantly and was relatively similar in both media, with a magnitude of about 23% (DW) and 30% (PBS), respectively. Thus, the null hypothesis of this study was rejected. 

Duration of specimen storage was previously reported to influence the outcome of POBS tests [3]. However, according to Collares et al., specimen storage of seven days compared to longer periods did not influence the outcome of POBS tests [3]. Unfortunately, only two studies with specimen storage of several months were included [3]. These two studies compared the results obtained for methacrylate resin-based sealers [16,17] with several other types of sealers, which in fact calls the results presented by Collares et al. [3] into question. Nonetheless, a significant influence of storage duration on the outcome of POBS tests was found with a threshold set at seven days of specimen storage, irrespective of the sealer type [3]. Only two studies are currently available comparing the POBS after different storage periods. In contrast to the present study, no statistically significant differences of the POBS values of AH Plus after two weeks or three months of incubation at 100% air humidity were reported [8]. In the second study, the POBS of AH Plus and a calcium-silicate sealer (EndoSequence BC (Brasseler USA, Savannah, GA, USA)) increased and were significantly higher after four weeks of incubation at 100% air humidity, compared to storage for only one week [18]. However, gutta-percha was used as a core material in both studies [8,18], whereas in the present investigation, canals were obturated with the sealer only. In conclusion, sparse data is available to compare the methodology of previous studies with that of the present study.

As AH Plus was reported to be nearly insoluble in DW or PBS [19], solubility can be eliminated as a possible explanation for the observed decreasing POBS values. Epoxy resin sealers adhere to root canal dentine via a covalent bond to dentinal collagen [20]. Therefore, degrading processes at the covalent junction of epoxy resin root canal sealers to root dentine after prolonged immersion in aqueous solutions may be a possible explanation for the present findings.

After immersion of BioRoot RCS in PBS or DW for one week, no significant differences in the POBS of BioRoot RCS were found. The formation of a mineral infiltration zone at the calcium silicate-dentine interface was reported as a possible mechanism of adhesion of calcium silicate cements to root dentine [21]. The adhesion of calcium silicate sealers via a mineral infiltration zone and dentinal tubule tags [21] therefore seems not to be affected by the type of immersion solution. BioRoot RCS was reported for prolonged setting time up to 11 h when stored in physiological media (HBSS or DMEM), compared to immersion in DW (approx. 4 h) [10]. Nonetheless, this fact may not have affected the dislodgement resistance after seven days of immersion [10]. The decrease of the POBS over the eight week period of BioRoot RCS in the present study was more substantial when specimens were stored in PBS (a decrease of POBS of 72%) compared to DW (a decrease of POBS of 28%). Solubility and sorption of BioRoot RCS were reported to increase when immersed in HBSS or DMEM compared to DW [10]. Physiological ion concentrations of the immersion medium obviously enhance the solubility of calcium silicate materials. Therefore, increased sorption and solubility of BioRoot RCS in PBS compared to DW could serve as an explanation for the inferior POBS of BioRoot RCS when stored in PBS for eight weeks. 

Irrespective of the results obtained for AH Plus and BioRoot RCS, the impact of the type of immersion solution used during the storing and the duration of the storing period on the push-out testing protocol seems to be a highly relevant result of the present study. Whilst other studies have already evaluated some variables in the POBS testing protocol [1,4,5,6,7,22], literature concerning this aspect is still lacking. The results of this study clearly reveal that different sealer types react significantly differently under varying experimental conditions. Therefore, the laboratory assessment of the POBS has to be considered incomparable when the environmental conditions of specimen storage are altered. Furthermore, the differing results of sealer types in combinations with media simulating physiological ion concentrations question the results obtained in DW or under humid air environments. Only in vitro studies conducted under conditions close to the clinical situation allow an interpretation of the results in reference to clinical situations. It is obvious that calcium silicate-based materials, in particular, are to be investigated in a manner simulating physiological environments in order to retain reliable and clinically applicable results.

A possible limitation of the experimental setup of the present study is the fact that dentine conditions and the micro- and macro-anatomy of the included teeth can only be standardized to a certain extent. In order to standardize the root specimens as well as possible, only single-rooted teeth were included and root slices were taken only from the mid-root section in order to ensure a similar number and size of dentinal tubules for all included specimens. Certainly, it would be preferable to distribute the dental slices of the same dental element in each of the subgroups in order to eliminate teeth condition bias, but this approach was not realizable. Eight groups were established in this study, and it is virtually impossible to obtain eight slices per tooth, all taken from the mid-root section. In order to compensate for this limitation, a markedly higher sample size (n = 60) was used than is usually utilized in comparable laboratory studies.

## 4. Materials and Methods

In the study, human single-rooted teeth with only one straight (curvature < 5°) root canal were included, which was checked by viewing their buccal and proximal radiographs using imaging software (ImageJ, NIH, Bethesda, MD, USA) as described previously [15]. The teeth were freshly extracted and stored in distilled water at 4 °C until further processing (storage was no longer than seven days) in order to avoid damage of the tooth structures. To exclude cracks, all roots were observed with a stereomicroscope under 20× magnification (Expert DN, Müller Optronic, Erfurt, Germany). The working length was obtained by measuring the length of the initial instrument (K-file ISO 10; VDW, Munich, Germany) at the major apical foramen minus 1 mm. All teeth were cut in a way that a working length of 18 mm was established. Patency of the canal was determined with K-files ISO 10 (VDW). Only teeth whose canal width near the terminus was approximately compatible with ISO 15 were included. This was checked with silver point sizes 10 and 15 (VDW). All root canals were instrumented with Gates-Glidden drills (Dentsply Sirona) up to size 6, resulting in a canal diameter of 1.5 mm, using the electric motor VDW.Gold (VDW). After each instrument, the root canal was irrigated with 5 mL of 3% NaOCl. 

Sample size calculation was performed using G*Power 3.1 (Heinrich Heine University, Düsseldorf, Germany) and resulted in a minimum of 239 samples (f = 0.25; α = 0.05; Power = 0.80). To reach a power of 0.99 (f = 0.25; α = 0.05), 480 samples were chosen.

The roots were embedded into acrylic resin vertically (Technovit 4071, Heraeus, Hanau, Germany) and sectioned horizontally with a 0.25 mm low-speed saw (Leitz, Wetzlar, Germany), beginning with a distance of 7 mm from the apex under constant water-cooling. Four slices of 1 mm thickness were obtained, representing the middle-portion of the root. In the case where the root canal preparation did not present as round in each root slice of a set, the set of four slices was discarded. Ultimately, 120 roots were chosen, resulting in a total of 480 specimens. The specimens were divided into eight groups of 15 roots each. The final irrigation protocol was immersion in 17% ethylenediaminetetraacetic acid (EDTA) for 60 seconds and immersion in DW twice for 30 seconds each. This irrigation protocol was performed to remove the smear layer completely and to avoid any influence of remnants of EDTA on the root canal surface on the outcome of the POBS test [15]. Finally, the canals were dried with paper points. 

The canals were obturated with either AH Plus or BioRoot RCS without the use of any core material (n = 240), as suggested by Oliveira et al. [13], as shown in Figure 2. 

The sealers were mixed according to the manufacturer’s instructions. The root specimens were fixed on panels covered with plastic foil to avoid an excess of sealer material. Sealers were introduced using a 1 mm spatula (STOMA, Emmingen-Liptingen, Germany). Eight groups (n = 60) of specimens were incubated at 37 °C covered with gauze moistened in DW or phosphate-buffered saline (PBS) for either one or eight weeks, resulting in the eight group distribution as shown in Figure 3. 

Complete setting of the sealers under these conditions was verified in a pilot study. All treatment procedures were carried out by the same operator, who was proficient in the techniques and materials used.

The specimens were placed in a metallic jig with a hole underneath. A standard size plunger with a tip of 1.2 mm in diameter was used to apply the vertical load onto the sealer surface. The diameter of the plunger tip was dimensioned with about 75% to 85% of the root canal diameter to ensure an equal distribution of the load on the sealer surface [1,5]. The vertical load in an apical-to-coronal direction was generated by a universal testing machine (Lloyd LF Plus/Nexygen, Ametek, Berwyn, USA) at a speed of 1 mm/min. The failure load was recorded in Newton when an abrupt reduction of the load was measured.

The lateral surface of the root canal of each specimen was calculated by the cylinder surface formula M=2·π·r·h. The POBS of each specimen was then calculated and expressed in N/mm^2^ (equivalent to MPa).

Photographs of each specimen were taken with a laser microscope (VK-X100, Keyence, Osaka, Japan) under 4× magnification. The mode of failure was evaluated by two blinded operators in three categories: adhesive failure (no material left on canal wall), cohesive failure (material present on entire canal wall), and mixed failure (material in patches on canal wall), as shown in Figure 4.

Statistical analysis of the POBS values was performed using multifactor ANOVA in order to assess interactions of the two variables (duration of immersion and type of immersion) and the post-hoc Student–Newman–Keuls test (*p* < 0.05), as data were distributed normally (Kolmogorov–Smirnov test). 

## 5. Conclusions

The POBS of an epoxy resin and a calcium silicate-based sealer decreased after a longer incubation time in immersion solutions. Duration of immersion and type of immersion solution displayed a significant impact on the POBS. Results of push-out testing protocols must not be compared without the consideration of duration of incubation and type of immersion solution. In order to obtain clinically relevant results from POBS tests, immersion solutions simulating body fluids (e.g., PBS) should be used.

## Figures and Tables

**Figure 1 materials-12-02860-f001:**
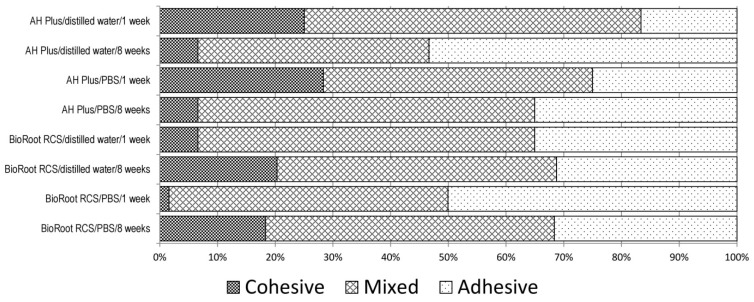
Percentage distribution of failure modes in all groups (%).

**Figure 2 materials-12-02860-f002:**
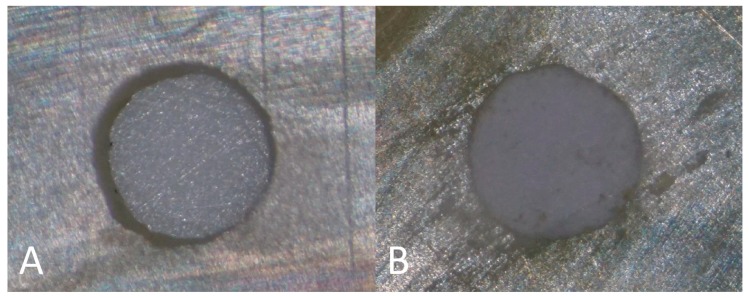
Images of the root slice with the prepared root canal before (**A**) and after (**B**) root canal obturation with BioRoot RCS.

**Figure 3 materials-12-02860-f003:**
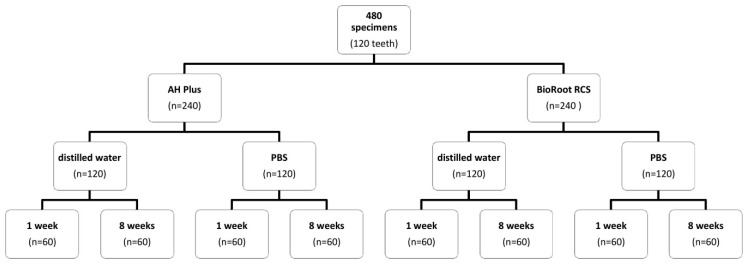
Specimen distribution into 8 groups (PBS: phosphate-buffered saline).

**Figure 4 materials-12-02860-f004:**
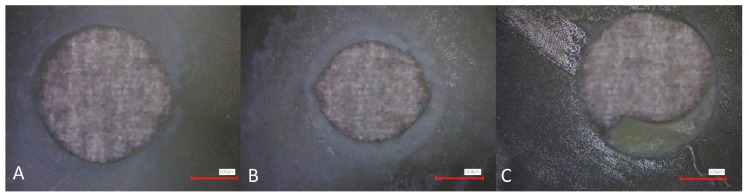
Images obtained by laser microscopy at 4× magnification for analysis of the mode of failure; examples for adhesive (**A**), cohesive (**B**), and mixed (**C**) failure types are given.

**Table 1 materials-12-02860-t001:** Push-out bond strength values according to sealer, immersion medium, and immersion period (N/mm^2^).

Group	Sealer	Immersion Medium	Immersion Period	Sample Size	Mean (N/mm^2^)	Standard Deviation
1	AH Plus	DW	1 week	60	2.64^b^	0.74
2			8 weeks	60	2.04^c^	0.85
3		PBS	1 week	60	3.75^a^	1.07
4			8 weeks	60	2.65^b^	1.10
5	BioRoot RCS	DW	1 week	60	1.80^c^	0.75
6			8 weeks	60	1.31^d^	0.71
7		PBS	1 week	60	1.98^c^	0.88
8			8 weeks	60	0.55^e^	0.58

Values with different superscripted letters indicate statistical differences at *p* = 0.05 (ANOVA and Student–Newman–Keuls test). DW: distilled water; PBS: phosphate-buffered saline.

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
