# Peer review of "Duration of Immersion and Type of Immersion Solution Distort the Outcome of Push-Out Bond Strength Testing Protocols"

_materials, 2019, doi:10.3390/ma12182860_

Round 1
Reviewer 1 Report
In the manuscript entitled: “Duration of immersion and type of immersion solution distort outcome of push-out bond strength testing protocols” the authors investigated the influence of duration of immersion and the type of immersion solution on the outcome of push-out bond strength (POBS) tests.
The authors found that AH Plus showed higher POBS when stored in PBS compared to DW irrespective of the incubation period. BioRoot RCS displayed higher POBS when stored in DW compared to PBS after 8 weeks of incubation. No difference was found after 1 week. Irrespective of the sealer or the immersion solution, POBS decreased from 1 week to 8 weeks. Mainly mixed failure modes were found in all groups irrespective of sealer, immersion medium and immersion period.
The authors concluded that POBS decreased after long-time incubation in both immersion solutions. Duration of immersion and type of immersion solution had a significant impact on the outcome of the POBS testing protocol.
Major comments:
In general, the idea and innovation of this study, regards the analysis of push-out bond strength testing protocols is interesting, because the management of this materialconditions is validated but further studies on this topic could be an innovative issue in this field could be open an innovative matter of debate in literature by adding new information. Moreover, there are few reports in the literature that studied this interesting topic with this kind of study design.
The study was well conducted by the authors; However, there are some concerns to revise that are described below.
The introduction section resumes the existing knowledge regarding the important factor linked with solution of in vitro analysis.
However, as the importance of the topic, the reviewer strongly recommends, before a further re-evaluation of the manuscript, to update the literature through read, discuss and cites in the references with great attention all of those recent interesting articles, that helps the authors to better introduce and discuss the aim of the study in light of the importance of microscopy and histology of and serum used for oral cancer analysis in dental field: 1) Kadić S, Baraba A, Miletić I, Ionescu A, Brambilla E, Ivanišević Malčić A, Gabrić D. Push-out bond strength of three different calcium silicate-based root-end filling materials after ultrasonic retrograde cavity preparation. Clin Oral Investig. 2018 Apr;22(3):1559-1565. doi: 10.1007/s00784-017-2244-6. 2) Ferlazzo N, Currò M, Zinellu A, Caccamo D, Isola G, Ventura V, Carru C, Matarese G, Ientile R. Influence of MTHFR Genetic Background on p16 and MGMT Methylation in Oral Squamous Cell Cancer. Int J Mol Sci. 2017 Mar 29;18(4). pii: E724. doi: 10.3390/ijms18040724.
The authors should be better specified, at the end of the introduction section, the rational of the study and the aim of the study with the null hypothesis. In the material and methods section, should better clarify how was performed the root canal obturation and the failure load. Moreover, specify if data were normalized or not. Please specify if was used a test unit.
The discussion section appears well organized with the relevant paper that support the conclusions, even if the authors should better discuss the importance of medium in the study in vitro in dentistry. The conclusion should reinforce in light of the discussions.
In conclusion, I am sure that the authors are fine clinicians who achieve very nice results with their adopted protocol. However, this study, in my view, does not in its current form, satisfy a very high scientific requirement for publication in this journal and requests a revision before publication.
Minor Comments:
Abstract:
Better formulate the introduction section by better describing the background
Introduction:
Please refer to major comments
Discussion
Please add a specific sentence that clarifies the results obtained in the first part of the discussion Page 4 last paragraph: Please reorganize this paragraph that is not clear
Author Response
Dear Reviewer,
thank you for your valuable comments. In following you find a point-by-point response. All necessary changes were made in the manuscript.
Major comments:
However, as the importance of the topic, the reviewer strongly recommends, before a further re-evaluation of the manuscript, to update the literature through read, discuss and cites in the references with great attention all of those recent interesting articles, that helps the authors to better introduce and discuss the aim of the study in light of the importance of microscopy and histology of and serum used for oral cancer analysis in dental field: 1) Kadić S, Baraba A, Miletić I, Ionescu A, Brambilla E, Ivanišević Malčić A, Gabrić D. Push-out bond strength of three different calcium silicate-based root-end filling materials after ultrasonic retrograde cavity preparation. Clin Oral Investig. 2018 Apr;22(3):1559-1565. doi: 10.1007/s00784-017-2244-6. 2) Ferlazzo N, Currò M, Zinellu A, Caccamo D, Isola G, Ventura V, Carru C, Matarese G, Ientile R. Influence of MTHFR Genetic Background on p16 and MGMT Methylation in Oral Squamous Cell Cancer. Int J Mol Sci. 2017 Mar 29;18(4). pii: E724. doi: 10.3390/ijms18040724.
We updated our literature research in the field of push-out bond strength of endodontic sealers. However, the studies of Kadic et al. and Ferlazzo et al. are not related to the field of endodontic sealers and unfortunately also not to our study.
The authors should be better specified, at the end of the introduction section, the rational of the study and the aim of the study with the null hypothesis.
This section was revised to clarify.
In the material and methods section, should better clarify how was performed the root canal obturation and the failure load.
This section was revised and found to be very detailed.
Moreover, specify if data were normalized or not.
This information is given at the end of the M&M section.
Please specify if was used a test unit.
As specified in the M&M, a universal testing machine (Lloyd LF Plus/Nexygen, Ametek, Berwyn, USA) was used.
The discussion section appears well organized with the relevant paper that support the conclusions, even if the authors should better discuss the importance of medium in the study in vitro in dentistry.
This is discussed in the last part of the discussion. We added additional information to clarify.
The conclusion should reinforce in light of the discussions.
This section was revised to clarify.
Minor Comments:
Abstract:
Better formulate the introduction section by better describing the background
The Abstract is already extended to the word limit.
Discussion
Please add a specific sentence that clarifies the results obtained in the first part of the discussion
Revised according to the comment.
Page 4 last paragraph: Please reorganize this paragraph that is not clear
This section was revised to clarify.
Reviewer 2 Report
Author names and affiliations need to be added to the manuscript
Remove the words “background”, “methods”, “results” and “conclusion” from the abstract. Please see a published paper in Materials for guidance
Line 33: please add references at the end of the phrase
Line 39: put ratio in italic
Please place the material and methods section after the introduction
How long did the teeth were stored before used in experiments? If they were stored immersed, which liquid was used? Did the authors think it can have affected the results?
The authors refer that “only teeth whose canal width near the terminus was approximately compatible with ISO 15 were included” and that “in cases, the root canal preparation did not present round in each root slice of a set, the set of four slices was discarded”. However, the authors refer they used 480 specimens from 120 teeth. If some teeth and specimens were discarded the final specimens’ number should be less. Please explain.
Line 179: The authors refer randomization before the roots were sectioned. Why the randomization occurred in this point and not only after the sections were obtained? Do authors think it would be better to do the randomization after the sections were obtained? Please explain and discuss if this can affect the observed results.
Line 179: how was randomization performed? Please explain.
In my opinion, if images of the tooth slices obtained and the filling with the cement could be added to the manuscript, it would increase readers interest.
Lines 184-186: Why the final irrigation protocol was chosen? Please explain.
How was the sample calculated? Did the authors perform a sample calculation test? Please explain.
Figures must be placed in the text and figure 2 must be renamed as figure 1
Figures must be placed in the text and figure 1 must be renamed as figure 2
Lines 206-208: microphotographs representative of each type of failure should be added to the manuscript
“p” value referring to statistical should be changed in all manuscript from P to “p”
Discussion section: a paragraph discussing the study limitations must be added to the manuscript
Author Response
Dear Reviewer,
thank you for your valuable comments. In following you find a point-by-point response. All necessary changes were made in the manuscript.
Author names and affiliations need to be added to the manuscript
The authors and affiliations were uploaded in our manuscript but removed by the editorial office.
Remove the words “background”, “methods”, “results” and “conclusion” from the abstract. Please see a published paper in Materials for guidance
Revised according to the comment.
Line 33: please add references at the end of the phrase
Revised according to the comment.
Line 39: put ratio in italic
Revised according to the comment.
Please place the material and methods section after the introduction
The order Introduction-Results-Discussion-M&M is predefined in the author instructions.
How long did the teeth were stored before used in experiments? If they were stored immersed, which liquid was used? Did the authors think it can have affected the results?
Information was added to the manuscript.
The authors refer that “only teeth whose canal width near the terminus was approximately compatible with ISO 15 were included” and that “in cases, the root canal preparation did not present round in each root slice of a set, the set of four slices was discarded”. However, the authors refer they used 480 specimens from 120 teeth. If some teeth and specimens were discarded the final specimens’ number should be less. Please explain.
Specimens were discarded after sectioning before further processing until a number of 120 teeth were established. This part was revised to clarify.
Line 179: The authors refer randomization before the roots were sectioned. Why the randomization occurred in this point and not only after the sections were obtained? Do authors think it would be better to do the randomization after the sections were obtained? Please explain and discuss if this can affect the observed results.
Please see comment above.
Line 179: how was randomization performed? Please explain.
Teeth were randomly divided into the groups. As no different characterizations of the specimens could be measured, no randomization process was possible.
In my opinion, if images of the tooth slices obtained and the filling with the cement could be added to the manuscript, it would increase readers interest.
Selected photos were added to the manuscript.
Lines 184-186: Why the final irrigation protocol was chosen? Please explain.
Information was added to the manuscript.
How was the sample calculated? Did the authors perform a sample calculation test? Please explain.
Sample size calcualtation was performed using g*power. Information was added to the manuscript.
Figures must be placed in the text and figure 2 must be renamed as figure 1
Revised according to the comment.
Lines 206-208: microphotographs representative of each type of failure should be added to the manuscript
Revised according to the comment.
“p” value referring to statistical should be changed in all manuscript from P to “p”
Revised according to the comment.
Discussion section: a paragraph discussing the study limitations must be added to the manuscript
Information was added to the manuscript.
Reviewer 3 Report
The manuscript materials-587362, titled “Duration of immersion and type of immersion solution distort outcome of push-out bond strength testing protocols” shows original aspects and presents interesting results, which need to be supported by microscope photos of the samples and images of your work, like sample preparing etc. However, better organizing the results chapter (there is only one table) would help better understanding.
The article has a scientific character, the references are relevant and the English is satisfactory. However some writing errors have to be corrected.
The authors and affiliations are missing.
Please prepare the abstract following the instructions for authors.
Please insert figures in the text.
Please modify Material and Methods as section 2, between Introduction and Results...
Line 35. Collares et al. (2016). Please remove year of publcation.
Line 37. "POBS values were influenced by country and year of publication". Is this correct?? Please check.
Line 42. "the use of bovine or human dentine". You already mentioned this in line 37.
Line 116. "No study, so far, compared the POBS after different immersion times and storage in different immersion mediums." Please pay more attention to this kind of statements. Are you 100% sure this is correct? Maybe, it would be better to declare that, to your knowledge (or a similar expression) no study, so far....
Line 146. "this fact did could not have". Please correct.
Line 174, 175, 177. K-files ISO 10 (VDW), silver points sizes 10 and 15 (VDW), VDW.Gold (VDW). Please give town and country as above, line 172.
Line 176. Gates-Glidden drills (Dentsply Sirona). Please give town and country of the manufacturer.
Line 206-209. You refer to specimen photos taken with a laser microscope and three types of failure. You should, by all means, insert selected photos in your article, to prove and document your work.
Author Response
Dear Reviewer,
thank you for your valuable comments. In following you find a point-by-point response. All necessary changes were made in the manuscript.
The manuscript materials-587362, titled “Duration of immersion and type of immersion solution distort outcome of push-out bond strength testing protocols” shows original aspects and presents interesting results, which need to be supported by microscope photos of the samples and images of your work, like sample preparing etc. However, better organizing the results chapter (there is only one table) would help better understanding.
Revised according to the comment. The manuscript now contains 4 Figures and 1 Table.
The article has a scientific character, the references are relevant and the English is satisfactory. However some writing errors have to be corrected.
The authors and affiliations are missing.
The authors and affiliations were uploaded in our manuscript but removed by the editorial office.
Please prepare the abstract following the instructions for authors.
Revised according to the comment.
Please insert figures in the text.
Figures were inserted to the text.
Please modify Material and Methods as section 2, between Introduction and Results...
The order Introduction-Results-Discussion-M&M is predefined in the author instructions and is in accordance with the style of the journal.
Line 35. Collares et al. (2016). Please remove year of publication.
Revised according to the comment.
Line 37. "POBS values were influenced by country and year of publication". Is this correct?? Please check.
We checked it again. This is a finding of Collares et al. 2016
Line 42. "the use of bovine or human dentine". You already mentioned this in line 37.
It was mentioned again because it was supported by different citations. The sentence was rewritten to clarify.
Line 116. "No study, so far, compared the POBS after different immersion times and storage in different immersion mediums." Please pay more attention to this kind of statements. Are you 100% sure this is correct? Maybe, it would be better to declare that, to your knowledge (or a similar expression) no study, so far....
This statement was a result of our literature search. Nonetheless it is better to rewrite the sentence according to the comment.
Line 146. "this fact did could not have". Please correct.
Revised according to the comment.
Line 174, 175, 177. K-files ISO 10 (VDW), silver points sizes 10 and 15 (VDW), VDW.Gold (VDW). Please give town and country as above, line 172.
It is usual to mention town and country of the manufacturer only once with the first nomination. In the following only the manufacturer is mentioned.
Line 176. Gates-Glidden drills (Dentsply Sirona). Please give town and country of the manufacturer.
Please see comment above.
Line 206-209. You refer to specimen photos taken with a laser microscope and three types of failure. You should, by all means, insert selected photos in your article, to prove and document your work.
Selected photos were added to the manuscript.
Round 2
Reviewer 1 Report
In the R1 version of the manuscript entitled: “Duration of immersion and type of immersion solution distort outcome of push-out bond strength testing protocols” the authors followed all the issues suggested by the reviewer. Though the changes based on the reviewer comments, almost of the criticisms were carefully analysed and solved.
I have carefully evaluated all parts of the manuscript. I believe that the article, in this version, is now adequate for publication in this journal.
Author Response
Thank you. Kind regards.
Reviewer 2 Report
Line 12: put per in italic
Please place the material and methods section after the introduction. Although it might be referred in instructions to authors otherwise, see published papers to confirm the order is not correct. Also, change the number of figures accordingly.
Lines 98-99: did the authors perform statistical analysis regarding the type of failure mode?
Line 206: please remove the word “randomly” from the phrase. If, as the authors stated in response do reviewers, no randomization process was possible, then it must not be referred that way in the text, since it induces readers in error Author Response
Dear Reviewer,
thank you for your valuable comments. In following you find a point-by-point response. All necessary changes were made in the manuscript.
Line 12: put per in italic
Revised according to the comment.
Please place the material and methods section after the introduction. Although it might be referred in instructions to authors otherwise, see published papers to confirm the order is not correct. Also, change the number of figures accordingly.
I will contact the editorial office about this matter.
We prepared the manuscript as demanded. We also checked the latest manuscripts published on the website and found different orders and structures. Please find this copy from the instructions for authors:
Manuscript Preparation
General Considerations
Research manuscripts should comprise: Front matter: Title, Author list, Affiliations, Abstract, Keywords Research manuscript sections: Introduction, Results, Discussion, Materials and Methods, Conclusions (optional).Back matter: Supplementary Materials, Acknowledgments, Author Contributions, Conflicts of Interest, References.
Lines 98-99: did the authors perform statistical analysis regarding the type of failure mode?
No, only descriptional analysis.
Line 206: please remove the word “randomly” from the phrase. If, as the authors stated in response do reviewers, no randomization process was possible, then it must not be referred that way in the text, since it induces readers in error
Revised according to the comment.
Reviewer 3 Report
Authors and affiliation still missing. Please clarify this matter with the editorial office. This did not happen in the past.
As suggested by 2 reviewers, please consult the template and follow the order Introduction-M&M-Results-Discussion-Conclusion.
Author Response
Dear Reviewer, thank you for your valuable comments. In following you find a point-by-point response.
Authors and affiliation still missing. Please clarify this matter with the editorial office. This did not happen in the past.
We will do so.
As suggested by 2 reviewers, please consult the template and follow the order Introduction-M&M-Results-Discussion-Conclusion.
I will contact the editorial office about this matter.
We prepared the manuscript as demanded. We also checked the latest manuscripts published on the website and found different orders and structures. Please find this copy from the instructions for authors:
Manuscript Preparation
General Considerations
Research manuscripts should comprise: Front matter: Title, Author list, Affiliations, Abstract, Keywords Research manuscript sections: Introduction, Results, Discussion, Materials and Methods, Conclusions (optional).Back matter: Supplementary Materials, Acknowledgments, Author Contributions, Conflicts of Interest, References.